# Increasing the Legume–Rhizobia Symbiotic Efficiency Due to the Synergy between Commercial Strains and Strains Isolated from Relict Symbiotic Systems

Vera Safronova [1,*], Anna Sazanova [1], Irina Kuznetsova [1], Andrey Belimov [1], Polina Guro [1], Denis Karlov [1], Oleg Yuzikhin [1], Elizaveta Chirak [1], Alla Verkhozina [2], Alexey Afonin [1], Evgeny Andronov [1,3] and Igor Tikhonovich [1,3]

1   All-Russia Research Institute for Agricultural Microbiology (ARRIAM), Sh. Podbelskogo 3, 196608 Saint Petersburg, Russia; anna_sazanova@mail.ru (A.S.); kuznetsova_rina@mail.ru (I.K.); belimov@rambler.ru (A.B.); polinaguro@gmail.com (P.G.); makondo07@gmail.com (D.K.); yuzikhin@gmail.com (O.Y.); chirak.elizaveta@gmail.com (E.C.); afoninalexeym@gmail.com (A.A.); eeandr@gmail.com (E.A.); arriam2008@yandex.ru (I.T.)
2   Siberian Institute of Plant Physiology and Biochemistry (SIPPB), 664033 Irkutsk, Russia; allaverh@list.ru
3   Department of Genetics and Biotechnology, Saint Petersburg State University, Universitetskaya Emb. 7/9, 199034 Saint Petersburg, Russia
*   Correspondence: v.safronova@rambler.ru; Tel.: +7-(812)470-51-00

**Abstract:** The phenomenon of rhizobial synergy was investigated to increase the efficiency of nitrogen-fixing symbiosis of alfalfa (*Medicago varia* Martyn), common vetch (*Vicia sativa* L.) or red clover (*Trifolium pratense* L.). These plants were co-inoculated with the respective commercial strains *Sinorhizobium meliloti* RCAM1750, *Rhizobium leguminosarum* RCAM0626 or *R. leguminosarum* RCAM1365 and with the strains *Mesorhizobium japonicum* Opo-235, *M. japonicum* Opo-242, *Bradyrhizobium* sp. Opo-243 or *M. kowhaii* Ach-343 isolated from the relict legumes *Oxytropis popoviana* Peschkova and *Astragalus chorinensis* Bunge. The isolates mentioned above had additional symbiotic genes (*fix, nif, nod, noe* and *nol*) as well as the genes promoting plant growth and symbiosis formation (*acdRS*, genes associated with the biosynthesis of gibberellins and auxins, genes of T3SS, T4SS and T6SS secretion systems) compared to the commercial strains. Nodulation assays showed that in some variants of co-inoculation the symbiotic parameters of plants such as nodule number, plant biomass or acetylene reduction activity were increased. We assume that the study of microbial synergy using rhizobia of relict legumes will make it possible to carry out targeted selection of co-microsymbionts to increase the efficiency of agricultural legume–rhizobia systems.

**Keywords:** plant-microbe interaction; crop legumes; relict symbiotic systems; rhizobial synergy; efficiency of symbiosis

## 1. Introduction

Legumes play an important role in natural ecosystems due to their ability to form symbiosis with nitrogen-fixing nodule bacteria (rhizobia). The formation of a mutually beneficial symbiosis allows the plant-microbial system to acquire new adaptive properties that increase its productivity and resistance to stress, as well as to ensure successful competition for ecological space. During co-evolution of leguminous plants and root nodule bacteria, the signaling system of interaction between symbionts arose, providing specific recognition and integration between partners. Knowledge about the mechanisms of specificity and evolution of symbiosis is necessary for science-based selection and design of highly effective plant-microbe agricultural systems in which the potential productivity of the plant is most realized. However, the complex mechanisms underlying the specificity of legume–rhizobia interactions and their optimization are still poorly understood, including

due to the lack of adequate experimental models. In this respect the symbiotic systems based on the relict leguminous plants may be of particular importance.

Earlier a representative collection of microsymbionts of Miocene–Pliocene relict legumes *Vavilovia formosa* (Steven) Fed., *Oxytropis triphylla* (Pall.) DC., *O. popoviana* Peschkova, *Astragalus chorinensis* Bunge and *Caragana jubata* (Pall.) Poir. originating from the North Caucasus and the Baikal Lake region (Russia), as well as from the Khuvsgul Lake region in Mongolia [1–5] was created. A large taxonomic diversity of the collected microsymbionts was demonstrated, including representatives of the families *Phyllobacteriaceae*, *Rhizobiaceae* and *Bradyrhizobiaceae*. Using these microsymbionts, the phenomenon of rhizobial synergy was discovered, which was expressed in the ability of strains with different taxonomic position and complementary sets of symbiotic genes to simultaneously be present in nodules and increase the rate of nodulation, number of nodules, nitrogen-fixing activity and plant biomass [3,4]. It was also shown that strains isolated from relict symbiotic systems may have a wide range of host plants, including symbiotically narrow-specific species of *Medicago sativa* L. and *Trifolium pratense* L. [4]. These results suggested that the symbiotic systems of relict plants are formed with various groups of rhizobia, which can be co-microsymbionts and jointly participate in increasing the effectiveness of symbiosis with a wide range of host plants, including agricultural crops.

Thus, symbiotic systems of relict legumes are promising models for studying the evolution of host specificity, the impact of natural rhizobial diversity on plant growth and multipartite interactions between symbiotic partners. In addition, the discovered phenomenon of rhizobial synergy can be used as a new principle to increase the productivity of traditional crop legumes. In this regard, the aim of this work was to study the effectiveness of symbiosis upon joint inoculation of alfalfa (*Medicago varia* Martyn), common vetch (*Vicia sativa* L.) and red clover (*Trifolium pratense* L.) with their commercial strains and strains isolated from *O. popoviana* and *A. chorinensis*. Specific tasks were as follows: (1) To select, using a comparative whole-genome analysis, the isolates having additional genes involved in the formation of symbiosis as compared to the commercial strains *Sinorhizobium meliloti* RCAM1750 (alfalfa), *Rhizobium leguminosarum* RCAM0626 (common vetch) and *R. leguminosarum* RCAM1365 (red clover). Attention was paid to genes *fix, nif, nod, noe* and *nol*, as well as genes promoting plant growth and symbiosis formation (*acdRS*; genes associated with the biosynthesis of gibberellins and auxins; genes of T3SS, T4SS and T6SS secretion systems). (2) To study the effects of co-inoculation of alfalfa, common vetch and red clover with commercial strains and the selected isolates in the gnotobiotic plant nodulation assays.

## 2. Materials and Methods

### 2.1. Bacterial Material

Commercial strains of rhizobia: *Sinorhizobium meliloti* RCAM1750 nodulating alfalfa (*M. varia*), *Rhizobium leguminosarum* RCAM0626 nodulating common vetch (*V. sativa*) and *R. leguminosarum* RCAM1365 nodulating red clover (*T. pratense*) were used. The rhizobial strains *Mesorhizobium japonicum* Opo-235, *M. japonicum* Opo-242, *Bradyrhizobium* sp. Opo-243 and *M. kowhaii* Ach-343 were isolated from nodules of the relict legumes *O. popoviana* and *Astragalus chorinensis*. For this purpose, soil samples and seeds of *O. popoviana* and *A. chorinensis* were collected in the Republic of Buryatia (Baikal Lake region, Russia). Seeds were surface sterilized and scarified by treatment with 0.1% $HgCl_2$ for 10 min and then 5% NaOCl for 8 min, rinsed carefully with sterile tap water and germinated on filter paper in Petri dishes at 25 °C in the dark for 4 days. Seedlings were transferred to three sterile plastic pots (one pot with 5 seeds) containing 250 g of soil. Plants were cultivated for 60 days in the growth chamber with 50% relative humidity and four-level illumination/temperatures mode: night (dark, 18 °C, 8 h), morning (200 $\mu$mol m$^{-2}$ s$^{-1}$, 20 °C, 2 h), day (400 $\mu$mol m$^{-2}$ s$^{-1}$, 23 °C, 12 h), evening (200 $\mu$mol m$^{-2}$ s$^{-1}$, 20 °C, 2 h). Illumination was performed by L 36W/77 FLUORA lamps (Osram, Munich, Germany). Then roots of individual plants were removed from soil and washed with tap water. Strains of nodule bacteria were isolated from the obtained nodules by the standard method described

earlier [6] and were cultivated using modified yeast extract mannitol agar (YMA) [7] supplemented with 0.5% succinate (YMSA) [8]. Information about strains used in this work is given in Table 1. The genera *Sinorhizobium* and *Mesorhizobium* belong to fast-growing rhizobia (colonies on agar medium appear on 3–4 days), and the genus *Bradyrhizobium*—to slowly-growing rhizobia (colonies appear on 6–7 days). All strains were deposited in the Russian Collection of Agricultural Microorganisms (RCAM), registered in the WFCC-MIRCEN World Data Centre for Microorganisms under the number WDCM 966 [9] and stored at −80 °C in the automated Tube Store (Liconic Instruments, Mauren, Lichtenstein) as described previously [10].

**Table 1.** Used commercial strains of rhizobia and strains isolated from the relict legumes originated from Baikal Lake region.

| Host Plant | Strain/Isolate | Species of Rhizobia |
|:---:|:---:|:---:|
| | Commercial strain | |
| *Vicia sativa* | RCAM0626 | *Rhizobium leguminosarum* |
| *Trifolium pratense* | RCAM1365 | *R. leguminosarum* |
| *Medicago sativa* | RCAM1750 | *Sinorhizobium meliloti* |
| | Isolate | |
| | Opo-235 | *Mesorhizobium japonicum* |
| *Oxytropis popoviana* | Opo-242 | *M. japonicum* |
| | Opo-243 | *Bradyrhizobium* sp. |
| *Astragalus horinensis* | Ach-343 | *M. kowhaii* |

### 2.2. Whole Genome Sequencing of the Strains under Investigation

For the whole genome sequencing of three commercial rhizobial strains deposited in the RCAM collection (*S. meliloti* RCAM1750, *R. leguminosarum* RCAM0626 and *R. leguminosarum* RCAM1365) their genomic DNA was extracted using Genomic DNA Purification KIT (Thermo Fisher Scientific, EU) according to recommendation of manufacturer. Long-read whole genome sequencing was performed using a MinIon sequencer (Oxford Nanopore, England) of the Core Centrum "Genomic Technologies, Proteomics and Cell Biology" at the ARRIAM. The SQK-LSK109 Ligation Sequencing Kit with the EXP-NBD104 Native Barcoding Expansion 1–12 kit was used to prepare the library according to manufacturer instructions. The reads were basecalled and demultiplexed using the guppy_basecaller (v. 3.3.0). The Flye pipeline (v 2.6-release) was used to assemble the Nanopore reads [11]. The resulting assembly was corrected 4 times using Racon (v. 1.3.2) (https://github.com/lbcb-sci/racon, accessed on 22 March 2013) with the following modifiers (−m 8 −x −6 −g −8 −w 500), following with a single polish using the medaka (v 0.10.0) program with default parameters [12]. Search for genes in the assembled contigs was performed using the RAST annotation service [13]. All the genomes were assembled into circular replicons. Genome sequencing of the Baikal isolates *Mesorhizobium japonicum* Opo-235, *M. japonicum* Opo-242, *Bradyrhizobium* sp. Opo-243 and *M. kowhaii* Ach-343 was performed on a MiSeq genomic sequencer (Illumina, San Diego, CA, USA), as described earlier [3,4].

Search for homologs of the symbiotic *fix, nif, nod, noe* and *nol* genes as well as genes that promote plant growth (*acdRS*, gibberellin- and auxin-biosynthesis related) and genes of the T3SS, T4SS and T6SS secretion systems in annotated genomes was performed using CLC Genomics Workbench 7.5.1 software using local BLASTn and tBLASTx [14].

The whole genome sequences have been deposited to the NCBI GenBank database under accession numbers: CP050511-CP050513 for the strain *S. meliloti* RCAM1750; CP050553-CP050557 for the strain *R. leguminosarum* RCAM0626; CP050514-CP050519 for the strain *R. leguminosarum* RCAM1365; QKOD00000000 for the isolate *M. japonicum* Opo-235;

MZXX00000000 for the isolate *M. japonicum* Opo-242; MZXW00000000 for the isolate *Bradyrhizobium* sp. Opo-243 and MZXV00000000 for the isolate *M. kowhaii* Ach-343.

*2.3. Plant Nodulation Assays*

Seeds of alfalfa (*M. varia*), common vetch (*V. sativa*) and red clover (*T. pratense*) plants were surface sterilized by $H_2SO_4$ for 10 min, washed with sterile tap water and germinated on filter paper in Petri dishes at 25 °C in the dark for 4 days. Germinated seedlings were transferred to 50 mL glass test tubes (2 seedlings per test tube) which contained 10 mL of sterile agar medium of the following composition (g/L): $K_2HPO_4$ 1.0, $KH_2PO_4$ 0.25, $MgSO_4$ 1.0, $Ca_3(PO_4)_2$ 0.2, $FeSO_4$ 0.02, $H_3BO_3$ 0.005, $(NH_4)_2MoO_4$ 0.005, $ZnSO_4$ 0.005, $MnSO_4$ 0.002, agar for micropropagation of plants (Dia-m, Moscow, Russia) 5.0.

After planting to the test tubes, two-days old seedlings were inoculated with individual strains or with a pair of strains in the amount of $10^6$ cells of each strain per test tube (10 tubes per inoculation variant). The cell concentration in suspensions was determined by the SmartSpec Plus Spectrophotometer (BioRad, Hercules, CA, USA). A suspension of each strain was prepared in a liquid medium of the same composition as described above (0.5 mL of each suspension per tube). In all mono-inoculation treatments an additional 0.5 mL of the liquid medium per tube was added. The uninoculated plants were used as negative control. Plants were cultivated for 24 (*T. pratense*), 25 (*M. varia*) or 26 (*V. sativa*) days in the growth chamber with 50% relative humidity and four-level illumination/temperatures mode: night (dark, 18 °C, 8 h), morning (200 $\mu$mol m$^{-2}$ s$^{-1}$, 20 °C, 2 h), day (400 $\mu$mol m$^{-2}$ s$^{-1}$, 23 °C, 12 h), evening (200 $\mu$mol m$^{-2}$ s$^{-1}$, 20 °C, 2 h). Illumination was performed by L 36W/77 FLUORA lamps (Osram, Munich, Germany). At the end of one representative experiment the nitrogen fixation of nodules in each test tube was measured by the acetylene-reduction method [15] using a gas chromatograph GC-2014 (Shimadzu, Japan). Then nodules were counted, shoots and roots were separated at the root collar and the fresh biomass of shoots and roots was immediately determined by the analytical balance Pioneer PX224 (Ohaus, Parsippany, NJ, USA). The level of acetylene reduction activity per one nodule in a tube was calculated. Strains were re-isolated from the obtained nodules (5 nodules from different tubes for each co-inoculation variant) and identified by 16S rDNA sequencing as described earlier [16]. Pictures of roots and nodules were performed by the stereo microscope Stemi 508 (Carl Zeiss, Oberkochen, Germany). The data were processed using the software STATISTICA version 10. Fisher's least significant difference (LSD) test, standard deviation (SD) and standard error (SE) were used to evaluate differences between means.

## 3. Results and Discussion

*3.1. Whole Genome Sequencing of the Strains under Investigation*

The genomic features of the studied strains are presented in Table 2.

**Table 2.** Genomic features of the studied isolates and the commercial strains.

| Strain/Isolate | Species | Coverage | Size (Mb) | GC% | Genes |
|---|---|---|---|---|---|
| RCAM0626 | *R. leguminosarum* | 91.0× | 7.70 | 60.89 | 4 376 |
| RCAM1365 | *R. leguminosarum* | 79.0× | 7.29 | 60.75 | 6 409 |
| RCAM1750 | *S. meliloti* | 110.0× | 6.92 | 62.10 | 5 863 |
| Opo-235 | *M. japonicum* | 58.0× | 7.36 | 62.60 | 6 791 |
| Opo-242 | *M. japonicum* | 20.0× | 7.35 | 62.60 | 6 777 |
| Opo-243 | *Bradyrhizobium* sp. | 12.0× | 8.63 | 63.80 | 7 890 |
| Ach-343 | *M. kowhaii* | 14.0× | 8.06 | 62.30 | 7 375 |

The currently described symbiotic *fix, nif, nod, noe* and *nol* genes as well as genes that promote plant growth (*acdRS*, gibberellin- and auxin-biosynthesis related) and genes of the T3SS, T4SS and T6SS secretion systems involved in the formation of symbiosis

were searched throughout the whole genome sequences of three commercial rhizobial strains (*S. meliloti* RCAM1750; *R. leguminosarum* RCAM0626, RCAM1365) and four strains isolated from the relict legumes *O. popoviana* (*M. japonicum* Opo-235, *M. japonicum* Opo-242, *Bradyrhizobium* sp. Opo-243) and *A. chorinensis* (*M. kowhaii* Ach-343). Presence of symbiotic *fix, nif, nod, noe* and *nol* genes is shown in Table 3.

The common *nodABC* genes necessary for legume nodulation [17] were found in all commercial strains and only in three isolates: *M. japonicum* Opo-235, *M. japonicum* Opo-242 and *M. kowhaii* Ach-343. These strains possessed also *nif* and *fix* genes required for nitrogen fixation and forming clusters: *nifHDK* and *nifENB* genes encoding structural and catalytic components of the nitrogenase complex [18] as well as *fixABCX* genes participating in electron transfer to nitrogenase [19]. At the same time, some of *fixNOPQ* and *fixGHIS* genes participating in symbiotically essential oxidase complex [20] were not observed in the commercial strains: the strain *R. leguminosarum* RCAM1365 lacked *fixQHS*; the strains *R. leguminosarum* RCAM0626 and *S. meliloti* RCAM1750 lacked *fixS* and *fixH*, respectively (Table 3). The isolate *Bradyrhizobium* sp. Opo-243 had only one group of *fixKJLNOQPGHIS* genes, which formed a single cluster. All the studied strains possessed some *nol* and *noe* genes, as well as additional *nod, nif* and *fix* genes. It should be noted that the set of symbiotic genes for each strain was unique.

Presence of the genes related to plant growth promotion (*acdRS*, gibberellin- and auxin-biosynthesis related genes) is shown in Table 4.

It was repeatedly shown that the *acdS* gene of rhizobia encoding enzyme 1-aminocyclopropane-1-carboxylate (ACC) deaminase plays important role in nodulation process [21]. Particularly, pea plants inoculated with ACC deaminase minus mutants of *Rhizobium leguminosarum* bv. *viciae* 128C53K [22] and 1066S [23] showed decreased nodule number, nitrogenase activity and shoot biomass as compared with a wild type. The commercial strains *R. leguminosarum* RCAM0626 and RCAM1365 did not contain the *acdS* gene. At the same time, the strain *S. meliloti* RCAM1750, as well as the isolates *Bradyrhizobium* sp. Opo-243 and *M. kowhaii* Ach-343, had both *acdS* and *acdR* genes required for the synthesis of ACC deaminase and its regulation, respectively. The isolates *M. japonicum* Opo-235 and *M. japonicum* Opo-242 contained only *acdS* gene.

All isolates had significantly more genes associated with the biosynthesis of gibberellins and auxins compared to the commercial strains (Table 4). It should be noted that gibberellin- and auxin-biosynthesis related genes are commonly known as factors responsible for plant growth, the efficiency of plant–microbe interactions and symbiosis formation. For example, they are involved in the processes of nodule initiation and development [24–26].

The genes of T3SS, T4SS and T6SS secretion systems presented in the studied strains are shown in Table 5. The T3SS secretion system was scarcely present in either commercial or isolated strains by one *fli* gene in the isolates *M. japonicum* Opo-235 and *M. kowhaii* Ach-343. In contrast, all strains, with the exception of the *R. leguminosarum* RCAM0626, possessed multiple *vir* genes belonging to theT4SS playing an important role in the translocation of a wide range of virulence factors into the host cell [27,28]. No genes of the T6SS secretion system were found in the commercial strains or in the isolate *Bradyrhizobium* sp. Opo-243, while four remaining isolates had different sets of these genes (Table 5). Most of them (*icmF, tssABCEGJKL, tagFH, vgrG, vasAEFK, clpV1, impA, vipAB, sciN, dotU,* Vgr family, Hcp family, FHA domain) was presented in the isolate *M. kowhaii* Ach-343. The T6SS secretion system, like T3SS and T4SS, is involved in bacterial virulence due to secretion of broad classes of effectors proteins: cytotoxins; lysozymes; lipoproteins; factors of adherence to epithelial cells and penetrating system, pilus formation, conjugation, protein translocation and so on [29–32]. It is known that these secretion systems in rhizobia can affect the development of plant–microbe interactions and the host specificity of strains, as well as determine the number of nodules formed [33].

**Table 3.** Presence of the symbiotic genes *fix, nif, nod, noe* and *nol* in the commercial rhizobial strains (*R. leguminosarum* RCAM0626, RCAM1365; *S. meliloti* RCAM1750) and four strains isolated from the relict legumes *O. popoviana* and *A. chorinensis* (*M. japonicum* Opo-235, Opo-242; *Bradyrhizobium* sp. Opo-243 and *M. kowhaii* Ach-343).

| Genes | Isolates from the Relict Legumes | | | | Commercial Rhizobial Strains | | |
|---|---|---|---|---|---|---|---|
| | **Opo-235** | **Opo-242** | **Opo-243** | **Ach-343** | **RCAM0626** | **RCAM1365** | **RCAM1750** |
| Fix copy 1 | NOQPGHIS | NOQPGHIS | – | – | NOQP | – | NOQPIS |
| Fix copy 2 | – | – | – | – | LNOQPGHI | – | – |
| Fix cluster | NOQPGHISJLXCBA | NOQPGHISLJXCBA | KJLNOQPGHIS | XCBANOQPGHIS | ABCXLNOQPGHI | GINOPABCX | LNOPGIABCX |
| *fixBA* | – | – | + | – | + | + | – |
| *fixJ* | – | – | – | – | + | + | – |
| *fixK* | + | + | – | – | – | – | + |
| Nif cluster | TZBAWVSHDKENX | BATZWSUQHDKENX | – | TZBAWSQHDKENX | HDKENAB | HDKENABT | XHDKENBAT |
| *nifL* | – | – | + | – | + | + | – |
| *nifR* | – | – | – | – | + | – | + |
| *nifS* | – | – | + | – | + | + | – |
| *nifU* | + | – | – | – | + | + | + |
| Nod cluster | ZJICBADEFL | ZJICBADEFL | – | LEFDABCIJZG | JICBADEFLMNO | JICBADEFLMND | MGNDABCIJQPGEFHD |
| *nodG* | – | + | + | – | + | + | + |
| *nodM* | – | + | – | + | + | – | + |
| *nodN* | – | + | + | + | + | + | + |
| *nodP* | + | + | + | + | – | + | + |
| *nodQ* | + | + | + | + | + | + | + |
| *nodT* | + | + | + | – | + | + | – |
| *nodV* | – | – | – | – | – | – | + |
| *nodW* | + | – | – | – | + | + | + |
| *noeK* | + | + | – | – | + | + | – |
| *noeL* | – | – | + | – | + | + | – |
| *nolK* | – | – | + | – | – | – | – |
| *nolF* | – | – | – | – | + | + | – |
| *nolL* | + | + | – | – | + | + | – |
| *nolR* | – | – | – | – | – | + | + |

**Table 4.** Presence of the genes that promote plant growth in the commercial rhizobial strains (*R. leguminosarum* RCAM0626, RCAM1365; *S. meliloti* RCAM1750) and four strains isolated from the relict legumes *O. popoviana* and *A. chorinensis* (*M. japonicum* Opo-235, *M. japonicum* Opo-242; *Bradyrhizobium* sp. Opo-243 and *M. kowhaii* Ach-343).

| Features/Genes | Isolates from the Relict Legumes | | | | Commercial Rhizobial Strains | | |
|---|---|---|---|---|---|---|---|
| | Opo-235 | Opo-242 | Opo-243 | Ach-343 | RCAM0626 | RCAM1365 | RCAM1750 |
| ACC deaminase synthesis | | | | | | | |
| *acdS* | + | + | + | + | − | − | + |
| *acdR* | − | − | + | + | − | − | + |
| Auxin synthesis | | | | | | | |
| nitrile hydratase α subunit | + | + | + | + | + | + | + |
| nitrile hydratase β subunit | + | + | + | + | + | + | + |
| tryptophan synthase α subunit | + | + | + | + | − | − | − |
| tryptophan synthase β subunit | + | + | + | + | − | − | − |
| indole-3-acetamide (IAM) hydrolase (*IaaH*) | − | − | + | + | − | − | − |
| amine oxidase | + | + | + | + | − | − | − |
| Gibberellin synthesis | | | | | | | |
| *cpxP* (CYP112) | + | + | + | + | − | − | + |
| *cpxR* (CYP114) | + | + | + | + | − | − | + |
| *cpxU* | + | + | + | + | − | − | − |
| *ispA* | + | + | + | + | − | − | − |
| SDR family | + | + | + | − | − | − | − |
| CPS (copalyl diphosphate synthase) cluster | − | − | − | + | − | − | − |

**Table 5.** Presence of the secretion systems T3SS, T4SS and T6SS genes in the commercial rhizobial strains (*R. leguminosarum* RCAM0626, RCAM1365; *S. meliloti* RCAM1750) and four strains isolated from the relict legumes *O. popoviana* and *A. chorinensis*.*—commercial strain, **—isolate from the relict legumes.

| Bacterial Species | Strain/Isolate | Secretion System | | |
|---|---|---|---|---|
| | | T3SS | T4SS | T6SS |
| *R. leguminosarum* | RCAM0626 * | - | - | - |
| | RCAM1365 * | - | *virB1-6,8-11* | - |
| *S. meliloti* | RCAM1750 * | - | *virB1-6,8-11* | - |
| *M. japonicum* | Opo-235 ** | *fli* | *virB2-6,8-11* | *icmF, tssABCEGJKL, tagFH, vgrG, vasA, clpV*, Hcp family |
| *M. japonicum* | Opo-242 ** | - | *virB1-6,8-11* | *clpV1, tssABCDEFG, tagH, tssJKLM, tagF*, PAAR domain, Vgr family, *tagH, tssLM* |
| *Bradyrhizobium* sp. | Opo-243 ** | - | *virB1-6,8-11* | - |
| *M. kowhaii* | Ach-343 ** | *fli* | *virB2-11* | *icmF, tssABCEGJKL, tagFH, vgrG, vasAEFK, clpV1, impA, vipAB, sciN, dotU*, Vgr family, Hcp family, FHA domain |

### 3.2. Plant Nodulation Assays

The plant nodulation tests were performed on alfalfa, common vetch and red clover plants with the participation of the corresponding commercial strains and the isolates *M. japonicum* Opo-235, *M. japonicum* Opo-242, *Bradyrhizobium* sp. Opo-243 or *M. kowhaii* Ach-343. Mono- and co-inoculation of alfalfa plants showed that three out of four isolates (except for *M. kowhaii* Ach-343) led to a significant increase in the number of nodules in variants of joint inoculation with the commercial strain *S. meliloti* RCAM1750 (Table 6). In some cases, this was accompanied by an increase in the biomass of shoots and the total biomass of plants (roots and shoots). An enhanced total nitrogen-fixing activity was observed only when the strain *S. meliloti* RCAM1750 was co-inoculated with *Bradyrhizobium* sp. Opo-243 (Figure 1), although the level of activity calculated per one nodule did not differ from the

commercial strain (data not shown). After co-inoculation with *S. meliloti* RCAM1750 and *M. kowhaii* Ach-343, despite a significant decrease in nodule number and nitrogen-fixing activity, an increase in the biomass of roots, shoots and total plant was observed.

**Table 6.** Effect of mono- and co-inoculation of alfalfa with the commercial strain *S. meliloti* RCAM1750 and the isolates *M. japonicum* Opo-235, *M. japonicum* Opo-242, *Bradyrhizobium* sp. Opo-243 or *M. kowhaii* Ach-343 in the sterile plant nodulation assay. The data means ± standard errors of one representative experiment (*n* = 10). Different letters show significant differences between treatments (Fisher's LSD test, *p* < 0.05). ↑ and ↓—significant increase or decrease, respectively, between inoculation with the strain RCAM1750 and other treatments. FW stands for fresh weight.

| Treatment | Number of Nodules (Tube$^{-1}$) | Total Plant Biomass (mg FW Plant$^{-1}$) | Root Biomass (mg FW Plant$^{-1}$) | Shoot Biomass (mg FW Plant$^{-1}$) |
|---|---|---|---|---|
| Control without inoculation | 0 | 87.8 ± 6.6 a | 46.4 ± 6.0 a | 41.4 ± 1.4 b |
| RCAM1750 | 7.0 ± 0.6 b | 91.6 ± 3.9 a | 48.4 ± 2.8 a | 43.2 ± 1.8 b |
| RCAM1750 + Opo-235 | 15.3 ± 1.5 c ↑ | 112.4 ± 5.2 bc ↑ | 56.6 ± 6.2 ab | 55.8 ± 3.2 cd ↑ |
| RCAM1750 + Opo-242 | 18.7 ± 0.9 d ↑ | 95.5 ± 7.3 ab | 47.5 ± 4.6 a | 48.0 ± 3.9 bc |
| RCAM1750 + Opo-243 | 14.3 ± 0.9 c ↑ | 102.5 ± 8.0 abc | 49.6 ± 3.1 a | 52.9 ± 5.3 c ↑ |
| RCAM1750 + Ach-343 | 3.7 ± 0.7 a ↓ | 137.7 ± 7.4 d ↑ | 70.9 ± 4.7 c ↑ | 66.8 ± 5.2 d ↑ |
| Opo-235 | 0 | 104.2 ± 4.2 abc | 66.9 ± 2.9 bc ↑ | 37.3 ± 4.2 ab |
| Opo-242 | 0 | 94.9 ± 6.6 ab | 46.3 ± 1.6 a | 49.2 ± 6.5 bc |
| Opo-243 | 0 | 91.8 ± 4.9 a | 63.2 ± 5.5 bc ↑ | 28.6 ± 0.8 a ↓ |
| Ach-343 | 0 | 119.5 ± 9.2 cd ↑ | 73.1 ± 3.9 c ↑ | 46.4 ± 5.5 bc |

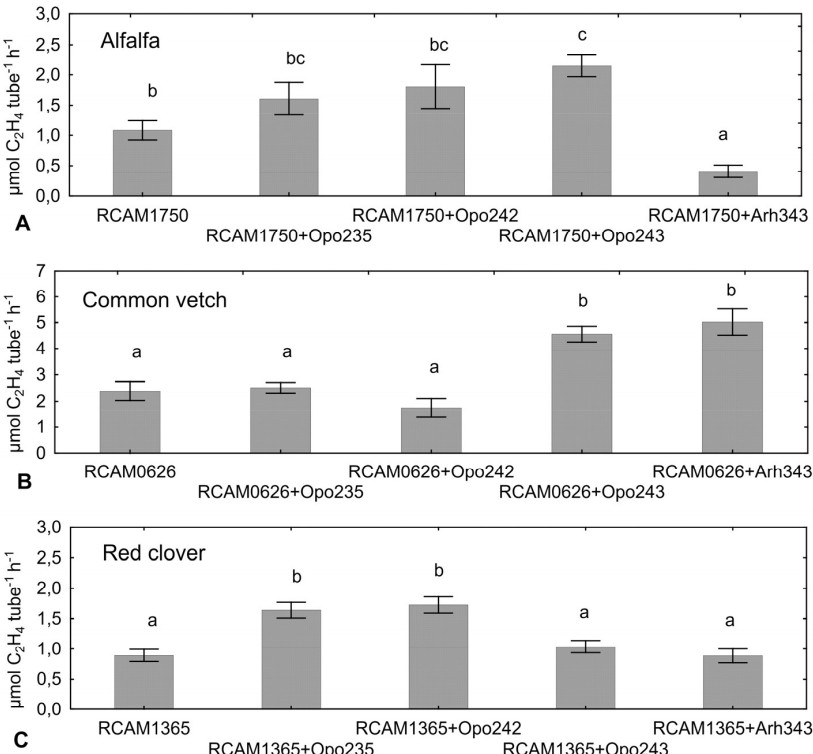

**Figure 1.** Acetylene reduction activity level in variants of mono- and co-inoculation of alfalfa (**A**), common vetch (**B**) and red clover (**C**) with the commercial strains *S. meliloti* RCAM1750, *R. leguminosarum* RCAM0626 or *R. leguminosarum* RCAM1365, respectively, and the isolates from the relict legumes. Different letters show significant differences between treatments (Fisher's LSD test, *p* < 0.05). Vertical bars represent SE.

The isolate *M. japonicum* Opo-235 co-inoculated with *S. meliloti* RCAM1750 contributed to significant acceleration of the nodule formation in comparison (Figure 2) and also led to the increase in plant biomass of alfalfa, although no increase in the total nitrogen-fixing activity was detected (Table 6, Figure 1).

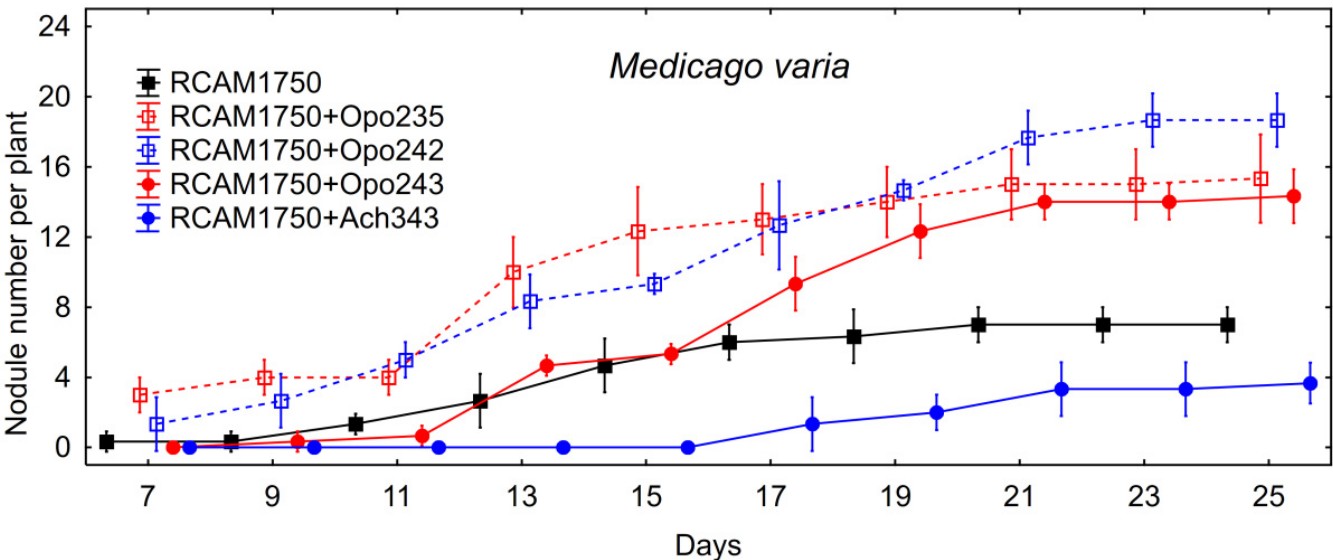

**Figure 2.** Nodule formation on alfalfa roots after mono- and co-inoculation with the commercial strain *S. meliloti* RCAM1750 and the isolates *M. japonicum* Opo-235, *M. japonicum* Opo-242, *Bradyrhizobium* sp. Opo-243 and *M. kowhaii* Ach-343 in the sterile plant nodulation assay. Vertical bars represent SD.

The other two isolates *M. japonicum* Opo-242 and *Bradyrhizobium* sp. Opo-243 also contributed to the formation of more nodules. However, *M. japonicum* Opo-242 had no effect on plant biomass, while *Bradyrhizobium* sp. Opo-243 increased the biomass of alfalfa shoots. It was shown that the nodules in the variant *S. meliloti* RCAM1750 + *M. japonicum* Opo-242 were small, and among them there were many tumor-like structures (Figure 3).

When the isolates *M. japonicum* Opo-235 and *Bradyrhizobium* sp. Opo-243 were used together with the strain *S. meliloti* RCAM1750, almost all nodules had a typical elongated shape, although both isolates themselves formed a small number of abnormal structures on the alfalfa roots (Figure 3). None of the isolates were detected in the alfalfa nodules and tumors either in mono-inoculation or in co-inoculation variants.

The data of plant nodulation experiment on mono- and co-inoculation of common vetch with the commercial strain *R. leguminosarum* RCAM0626 and the isolates are shown in the Table 7. Unlike alfalfa plants, the isolates themselves did not form nodules or tumors on the common vetch roots. Isolate *M. kowhaii* Ach-343 increased nodule number when co-inoculated with *R. leguminosarum* RCAM0626, while the isolate *M. japonicum* Opo-242 had no effect on this parameter. The level of total nitrogen-fixing activity enhanced upon co-inoculation with *R. leguminosarum* RCAM0626 and *Bradyrhizobium* sp. Opo-243 or *M. kowhaii* Ach-343 (Figure 1). However, this was associated with the formation of a significantly larger number of nodules, but not with their activity per one nodule (data not shown). An increase in the nodule number and the level of nitrogen-fixing activity had practically no effect on the plant biomass (Table 7). It is possible that some unknown growth limiting factors were presented under model gnotobiotic conditions and a potential of such symbiosis for biomass production could be more pronounced under optimal conditions. The study of the nodulation rate on the common vetch roots showed that only *Bradyrhizobium* sp. Opo-243 contributed to a significantly faster nodule formation compared to inoculation with *R. leguminosarum* RCAM0626 (Figure 4).

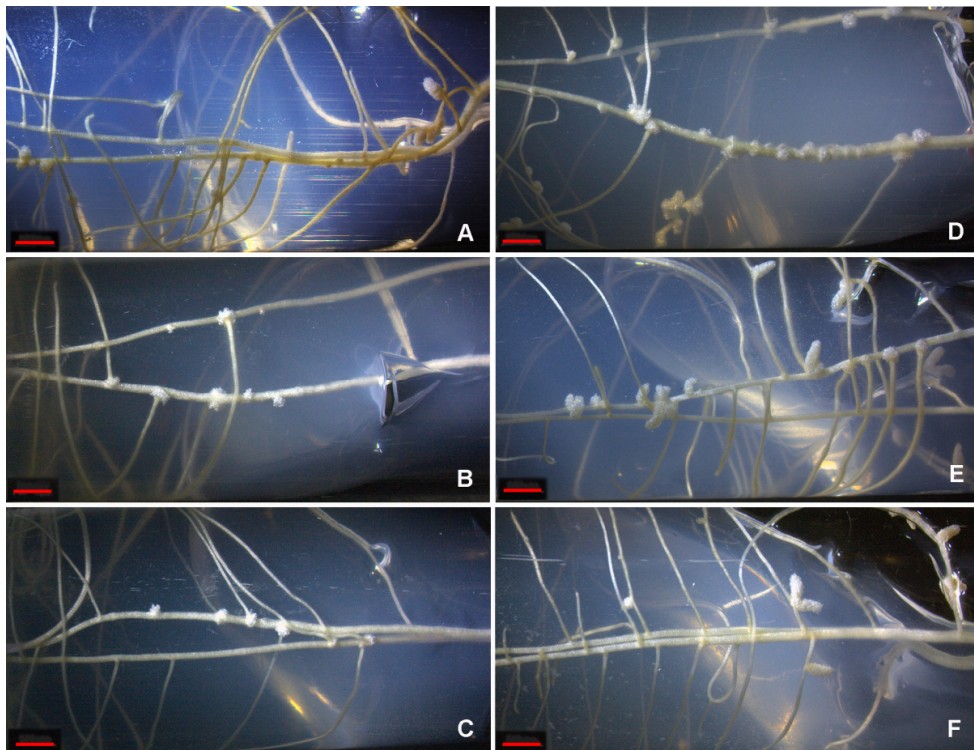

**Figure 3.** Pictures of the alfalfa roots in the sterile nodulation assay. Variants of mono-inoculation: (**A**) commercial strain *S. meliloti* RCAM1750, (**B**)—isolate *M. japonicum* Opo-235, (**C**) isolate *Bradyrhizobium* sp. Opo-243; variants of co-inoculation: (**D**) *S. meliloti* RCAM1750 + *M. japonicum* Opo-242, (**E**) *S. meliloti* RCAM1750 + *M. japonicum* Opo-235, (**F**) *S. meliloti* RCAM1750 + *Bradyrhizobium* sp. Opo-243. Scale bar = 0.5 mm.

**Table 7.** Effect of mono- and co-inoculation of common vetch with the commercial strain *R. leguminosarum* RCAM0626 and the isolates *M. japonicum* Opo-235, *M. japonicum* Opo-242, *Bradyrhizobium* sp. Opo-243 or *M. kowhaii* Ach-343 in the sterile plant nodulation assay. The data means ± standard errors of one representative experiment ($n = 10$). Different letters show significant differences between treatments (Fisher's LSD test, $p < 0.05$). ↑ and ↓—significant increase or decrease, respectively, between inoculation with the strain RCAM0626 and other treatments. FW stands for fresh weight.

| Treatment | Number of Nodules (Tube$^{-1}$) | Total Plant Biomass (mg FW Plant$^{-1}$) | Root Biomass (mg FW Plant$^{-1}$) | Shoot Biomass (mg FW Plant$^{-1}$) |
|---|---|---|---|---|
| Control without inoculation | 0 | 1152 ± 93 b | 672 ± 60 c | 480 ± 50 a |
| RCAM0626 | 52.7 ± 7.2 a | 1280 ± 112 b | 650 ± 93 c | 630 ± 41 bc |
| RCAM0626 + Opo-235 | 135.0 ± 6.4 c ↑ | 1232 ± 83 b | 673 ± 44 c | 559 ± 44 ab |
| RCAM0626 + Opo-242 | 49.3 ± 9.4 a | 1170 ± 158 b | 601 ± 72 bc | 599 ± 86 abc |
| RCAM0626 + Opo-243 | 107.0 ± 6.4 b ↑ | 1276 ± 130 b | 591 ± 68 abc | 685 ± 63 bc |
| RCAM0626 + Ach-343 | 95.3 ± 11.6 b ↑ | 1460 ± 127 bc | 793 ± 61 c | 667 ± 72 bc |
| Opo-235 | 0 | 1789 ± 159 c ↑ | 1047 ± 113 d ↑ | 742 ± 49 c |
| Opo-242 | 0 | 881 ± 153 a ↓ | 451 ± 101 ab ↓ | 430 ± 54 a ↓ |
| Opo-243 | 0 | 1240 ± 108 b | 683 ± 51 c | 568 ± 56 ab |
| Ach-343 | 0 | 853 ± 122 a ↓ | 382 ± 50 a ↓ | 471 ± 73 a ↓ |

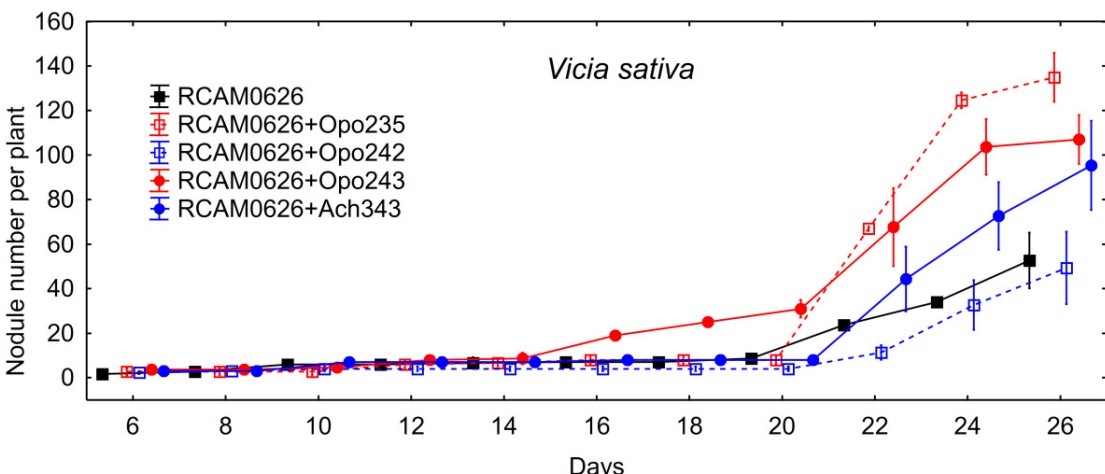

**Figure 4.** Nodule formation on common vetch roots after mono- and co-inoculation with the commercial strain *R. leguminosarum* RCAM0626 and the isolates in the sterile plant nodulation assay. Vertical bars represent SD.

In the experiment on *T. pratense*, the increase in the level of nitrogen-fixing activity was detected when *R. leguminosarum* RCAM1365 was combined with *M. japonicum* Opo-235 or *M. japonicum* Opo-242 (Figure 1). The level of activity calculated per one nodule in these two variants was also statistically significant increased (129.1 and 125.4 nmol $C_2H_4$ $nodule^{-1}$ $h^{-1}$, respectively), compared to mono-inoculation with the commercial strain RCAM1365 (66.2 nmol $C_2H_4$ $nodule^{-1}$ $h^{-1}$).

In this regard, it is important to note that in these two variants of co-inoculation the isolates were re-isolated from nodules (the data not presented). At the same time, *M. japonicum* Opo-235 itself formed a small number of typical elongated nodules on the *T. pratense* roots containing cells of this strain, whereas the isolate *M. japonicum* Opo-242 formed tumors (Figure 5) without bacteria inside them. Studying the nodulation rate on red clover did not reveal any variants of co-inoculation that differed from mono-inoculation with the commercial strain *R. leguminosarum* RCAM1365 (data not shown).

No change in the nodule number was observed as compared to the commercial strain (Table 8). It should be noted that the increase in the level of nitrogen-fixing activity did not lead to a significant increase in plant weight.

### 3.3. Comparison of the Genomic Data and Plant Nodulation Assays

We tried to explain the influence of the Baikal isolates on the efficiency of symbiosis with alfalfa, common vetch and red clover using a comparative genomic analysis of all strains that participated in plant nodulation experiments. Symbiotic and non-symbiotic genes that are absent in the commercial strains *S. meliloti* RCAM1750, *R. leguminosarum* RCAM0626 and *R. leguminosarum* RCAM1365 but are present in the isolates are shown in Table 9. Based on the analysis of the results obtained in the nodulation assay with alfalfa it can be assumed that the symbiotic genes *nodLZ, nolKL* and *noeKL* could act as the main accessory genes of the isolates, which contributed to an increase in the rate of nodulation and the efficiency of symbiosis with this plant species. The genes *nodL* and *nodZ* perform, respectively, 6-O-acetylation and fucosilation of the non-reducing end of the Nod factor (NF) and has been shown to be important for symbiosis of alfalfa and soybean with *Sinorhizobium* strains [34–37]. The *noeK, noeL,* and *nolK* genes associated with the expression of the *nodZ* gene are involved in the synthesis of the precursor of the fucosyl group from mannose [37–39]. The gene *nolL* carries out 4-O-acetylation of the fucose residue in NF, which arises as a result of the work of the gene *nodZ* [40]. Thus, all of the genes discussed above are responsible for the specificity of the plant–rhizobia interactions by NFs modification (acetylation and fucosilation). The *nodT* gene, also found in the isolates but not in *S. meliloti* RCAM1750 (Table 9), encodes an outer membrane lipoprotein playing a role in the secretion of NFs [41,42].

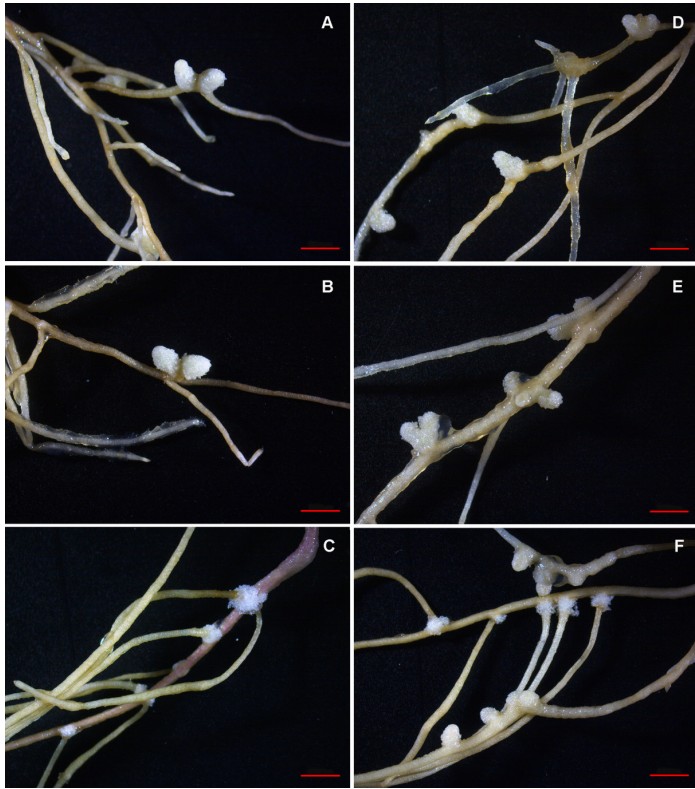

**Figure 5.** Pictures of red clover nodules in the sterile nodulation assay. Variants of mono-inoculation: (**A**) commercial strain *R. leguminosarum* RCAM1365, (**B**) isolate *M. japonicum* Opo-235, (**C**) isolate *M. japonicum* Opo-242; variants of co-inoculation: (**D**) *R. leguminosarum* RCAM1365 + *Bradyrhizobium* sp. Opo-243, (**E**) *R. leguminosarum* RCAM1365 + *M. japonicum* Opo-235, (**F**) *R. leguminosarum* RCAM1365 + *M. japonicum* Opo-242. Scale bar = 2.0 mm.

**Table 8.** Effect of mono- and co-inoculation of red clover with the commercial strain *R. leguminosarum* RCAM1365 and the isolates *M. japonicum* Opo-235, *M. japonicum* Opo-242, *Bradyrhizobium* sp. Opo-243 or *M. kowhaii* Ach-343 in the sterile plant nodulation assay. The data means ± standard errors of one representative experiment (*n* = 10). Different letters show significant differences between treatments (Fisher's LSD test, $p < 0.05$). ↑ and ↓—significant increase or decrease, respectively, between inoculation with the strain RCAM1365 and other treatments. FW stands for fresh weight.

| Treatment | Number of Nodules (Tube$^{-1}$) | Total Plant Biomass (mg FW Plant$^{-1}$) | Root Biomass (mg FW Plant$^{-1}$) | Shoot Biomass (mg FW Plant$^{-1}$) |
|---|---|---|---|---|
| Control without inoculation | 0 | 83.2 ± 9.8 ab | 39.0 ± 5.4 ab | 44.2 ± 4.7 a |
| RCAM1365 | 13.3 ± 2.0 a | 78.9 ± 5.9 a | 22.5 ± 2.1 a | 56.4 ± 7.9 abc |
| RCAM1365 + Opo-235 | 12.7 ± 2.0 a | 88.0 ± 8.3 abc | 28.2 ± 3.2 a | 59.8 ± 6.7 bcd |
| RCAM1365 + Opo-242 | 13.8 ± 2.0 a | 95.6 ± 9.1 abc | 22.8 ± 5.6 a | 72.8 ± 3.6 d ↑ |
| RCAM1365 + Opo-243 | 15.0 ± 1.7 a | 83.2 ± 11.4 ab | 27.4 ± 8.5 a | 55.8 ± 4.8 abc |
| RCAM1365 + Ach-343 | 15.7 ± 1.5 a | 96.3 ± 7.5 abc | 31.8 ± 4.5 a | 64.5 ± 3.2 cd |
| Opo-235 | 2.0 ± 0.3 b ↓ | 109.3 ± 0.4 c ↑ | 49.4 ± 2.7 bc ↑ | 59.9 ± 3.1 bcd |
| Opo-242 | 0 | 110.0 ± 8.2 c ↑ | 62.7 ± 9.3 c ↑ | 47.5 ± 1.1 ab |
| Opo-243 | 0 | 106.7 ± 9.5 bc ↑ | 58.7 ± 2.9 c ↑ | 48.0 ± 7.8 ab |
| Ach-343 | 0 | 108.5 ± 7.3 c ↑ | 55.2 ± 8.3 bc ↑ | 53.2 ± 4.7 abc |

**Table 9.** Symbiotic and non-symbiotic genes that are present in the isolates and absent in the commercial strains *S. meliloti* RCAM1750, *R. leguminosarum* RCAM0626 and *R. leguminosarum* RCAM1365. SG—symbiotic genes, NSG—non-symbiotic genes.

| Isolate | Commercial Strain | | | | | |
| | *S. meliloti* RCAM1750 | | *R. leguminosarum* RCAM0626 | | *R. leguminosarum* RCAM1365 | |
| | SG | NSG | SG | NSG | SG | NSG |
|---|---|---|---|---|---|---|
| *M. japonicum* Opo-235 | *nodTLZ, nolL, noeK* | T3SS, T6SS, *trp*, amine oxidase, *cpxU, ispA*, SDR family | *nodPZ, nifXTZVW, fixSK* | T4SS, T3SS, T6SS, *acdS, trp*, amine oxidase, *cpxPRU, ispA*, SDR family | *nodZ, nifXZVW, fixHSKLQ* | T3SS, T6SS, *acdS, trp*, amine oxidase, *cpxPRU, ispA*, SDR family |
| *M. japonicum* Opo-242 | *nodTLZ, nolL, noeK, nifQZWS, fixHJ* | T6SS, *trp*, amine oxidase, *cpxU, ispA*, SDR family | *nodPZ, nifQTZXW, fixSK* | T4SS, T6SS, *acdS, trp*, amine oxidase, *cpxPRU, ispA*, SDR family | *nodZ, nifQZXW, fixHSKLQ* | T6SS, *acdS, trp*, amine oxidase, *cpxPRU, ispA*, SDR family |
| *Bradyrhizobium* sp. Opo-243 | *nodT, nolK, noeL, nifLS, fixHJ* | *iaaH, trp*, amine oxidase, *cpxU, ispA*, SDR family | *nodP, nolK, fixSK* | T4SS, *acdRS, iaaH, trp*, amine oxidase, *cpxPRU, ispA*, SDR family | *nod, nolK, fixHSKLQ* | *acdRS, iaaH, trp*, amine oxidase, *cpxPRU, ispA*, SDR family |
| *M. kowhaii* Ach-343 | *nodLZ, nifQZWS, fixH* | T3SS, T6SS, *iaaH, trp*, amine oxidase, *cpxU, ispA*, CPS cluster | *nodPZ, nifQTZXW, fixS* | T4SS, T3SS, T6SS, *acdRS, iaaH, trp*, amine oxidase, *cpxPRU, ispA*, CPS cluster | *nodZ, nifQZXW, fixQHS* | T3SS, T6SS, *acdRS, iaaH, trp*, amine oxidase, *cpxPRU, ispA*, CPS cluster |

However, a positive effect of the gene *nodT* on the formation of symbiosis upon joint inoculation with the commercial strain seems unlikely. We also do not focus on the *nif* and *fix* genes of isolates that are absent in the strain *S. meliloti* RCAM1750, since none of the isolates was detected in the alfalfa nodules and tumors either in mono-inoculation or in co-inoculation variants.

Genes of the isolates that were absent in the commercial strain *R. leguminosarum* RCAM0626 are shown in Table 9. Among them, the *nodPZ* and *nolK* genes could affect the rate of common vetch nodule formation, their total number and/or nitrogen-fixing activity. The *nodP* gene is involved in the 6-O-sulfation of the reducing end of the NFs [17]. It was known that NodP functioned in conjunction with NodQ, which synthesizes the donor of the sulfate group [43]. It should be noted that the commercial strain *R. leguminosarum* RCAM0626 has the *nodQ* gene, although the importance of NFs sulfation for the common vetch has not been shown. The functions of *nodZ* and *nolK* genes have been described above. It is noteworthy that the *nolK* gene is present only in the isolate *Bradyrhizobium* sp. Opo-243, which alone contributed to the acceleration of nodulation on *V. sativa* plants (Figure 4). The *nodZ* and *nolK* genes were found by other authors in a number of *R. leguminosarum* strains, e.g., in many pea nodulating rhizobia the NFs carry a fucosyl group, which may indicate the importance of this modification for the tribe Fabeae [44]. Since we did not detect an increase in nitrogen-fixing activity per nodule in variants of co-inoculation of common vetch, we do not discuss the role of *nif* and *fix* genes that are present in isolates, although the strain *M. kowhaii* Ach-343 was re-isolated from the nodules together with the commercial strain RCAM0626.

The nodulation experiment with red clover, unlike those with alfalfa and common vetch, showed an increase in the level of nitrogen-fixing activity per one nodule caused by co-inoculations. The additional *nif* and *fix* genes of *M. japonicum* Opo-235 and *M. japonicum* Opo-242 might contributed to this parameter when co-inoculated with *R. leguminosarum* RCAM1365 (Table 9). Among them, *nifZQXW* genes associated with the synthesis of the Fe-S-cofactor, fixation of molybdenum and protection of nitrogenase from oxygen damage, as well as the *nifV* gene, which is involved in the maturation of nitrogenase through the synthesis of homocitrate [45–49]. The *fixHSQ* genes are involved in the synthesis of components of the oxidase complex, which is associated with the membrane and allows cells to breathe at low oxygen concentrations [20,46]. The *fixKL* genes, together with the

*fixJT* genes, determine the expression of genes necessary for growth under microaerophilic conditions [20,50].

It should be noted that, in our previous work devoted to the analysis of genetic complementation in strains isolated from the relict symbiotic systems, the genes indicated above have already come into view. Thus, the *nolK* gene was mentioned in connection with the ability of the isolate *Bradyrhizobium* sp. Opo-243 to increase the rate of nodulation on the relict legume *Oxytropis popoviana* (host plant) when used with its paired strain *M. japonicum* Opo-242 isolated from the same nodule [3]. The presence of the complementary genes *nifQV* and *fixJKL* in *M. japonicum* Opo-235 and *M. kowhaii* Ach-343 was associated with an increase in the nitrogen-fixing activity of nodules on *Glycyrrhiza uralensis* plants co-inoculated with both of these strains. Moreover, it was shown that isolates complementing each other for these genes were localized in the same nodule cells [4].

The genes responsible for the synthesis of auxins and genes of the T6SS secretion system can also positively influence the process of nodule formation. The presence of these genes in *M. kowhaii* Ach-343 apparently caused a significant increase in biomass of alfalfa plants with a decrease in the number of root nodules (Table 6). The influence of secretion systems on the formation of plant–microbial relationships was previously shown on the example of two strains (*M. loti* 582 and *M. huakuii* 583) isolated from the endemic of Kamchatka *Oxytropis kamtschatica* [16]. These strains possessed the T3SS and T6SS genes and formed two types of nodules (typical elongated and abnormal rounded) on roots of the host plant after mono-inoculations. It was suggested that this phenomenon can be regulated by two different nodulation strategies: (1) NF dependent, by the infection thread formation, and (2) NF independent, by the direct penetration of bacteria through the host cell wall. However, it certainly cannot be ruled out that other genetic factors, which are not considered here, could influence the process of the symbiosis formation.

The mechanisms of the rhizobial synergy described above, which are associated with the genetic complementation of co-microsymbionts, can be based on interbacterial interactions involving membrane and secreted proteins. In addition, some affecting proteins can enter the environment after cell lysis. The phenomenon of bacterial synergy in the context of accelerating and increasing the efficiency of nodulation is most easily explained by the modification of NFs that occurs outside the cells or on their surface. At the same time, the cooperation between co-microsymbionts most likely depends on their spatial arrangement (rhizosphere, rhizoplane or nodule), and in the case of localization of bacteria in one nodule or in the same plant cells, the level of interaction should be higher.

## 4. Conclusions

Thus, it was shown that the strains isolated from relict leguminous plants differ significantly in the set of symbiotic genes, as well as the genes related to other genetic systems that affect plant–microbe interactions. Moreover, the studied isolates obtained from relict legumes have a noticeably greater diversity of such genes as compared to the commercial strains. The cooperation of rhizobial strains simultaneously present either on roots or in nodules was expressed in a change (both positive and negative) in the efficiency of symbiosis upon joint inoculation with commercial strains. It is assumed that carrying out genomic analysis of rhizobia will allow selecting complementary rhizobial strains and assessing the contribution of various genetic loci (including those with little-known functions) to the formation of plant–microbial relationships. It could be expected that the study of microbial synergy using rhizobia of relict legumes will make it possible to carry out targeted selection of co-microsymbionts for the increasing of the efficiency of agricultural legume–rhizobia systems.

**Author Contributions:** Conceptualization and writing, V.S. and E.A.; methodology, I.K., E.C., A.A. and O.Y.; investigation, A.S., D.K. and P.G.; resources, A.B. and A.V.; supervision, I.T. All authors have read and agreed to the published version of the manuscript.

**Funding:** The article was made with support of the Ministry of Science and Higher Education of the Russian Federation in accordance with agreement № 075-15-2020-920 date 16 November 2020 on providing a grant in the form of subsidies from the Federal budget of Russian Federation. The grant was provided for state support for the creation and development of a World-class Scientific Center "Agrotechnologies for the Future".

**Data Availability Statement:** Not applicable.

**Conflicts of Interest:** The authors declare no conflict of interest. The funders had no role in the design of the study; in the collection, analyses, or interpretation of data; in the writing of the manuscript, or in the decision to publish the results.

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
