# Peer review of "Increasing the Legume–Rhizobia Symbiotic Efficiency Due to the Synergy between Commercial Strains and Strains Isolated from Relict Symbiotic Systems"

_agronomy, doi:10.3390/agronomy11071398_

Round 1
Reviewer 1 Report
Could you please specify how did you inoculate the plants with bacteria? Otherwise than this I have no other comments.
Reviewer 2 Report
This manuscript written by Safronova, V. et al explores “Increasing the legume-rhizobia symbiotic efficiency due to the synergy between commercial strains and strains isolated from 3 relict symbiotic systems”
- It would be better to discuss more the relict legume importance in this study.
- In figure 1 for nodulation assay, It is interesting to see the variation in nodule number according to different inoculation groups. However, it is important to see the growth pattern of each isolate. Does that also affect in nodulation in the root? How about the autoregulation of nodulation (AON) effect in the 25 days harvested plants?
- Authors have mentioned that “An increase in the nodule number and the 281 level of nitrogen-fixing activity had practically no effect on the plant biomass”, It would be interesting to see the effect of N-fixation (acetylene reduction assay) activity based on the nodule number of different isolates.
- The tables look very crowded/text heavy, I would recommend making some graphs as well for some tables for example table 5. Graphs make it way easier to grab overall information with statistics letters.
- The authors are not very clear about the statistical tool that they used for this experiment. Please elaborate on the tests that were used. Do all those data meet the criteria for the test?
- Overall, the paper can be published if all those comments have been addressed.
Reviewer 3 Report
The manuscript describes an interesting work in which the authors use four of their bacteria strains isolated from relict symbiotic systems together with commercial nitrogen-fixing symbiotic strains to see their possible effect on the N-fixing efficiencies on three different host plants. They have done WGS on the studied strains providing insight into their symbiotic gene set as well as other genes with interest such as those of secretion systems or plant growth promoting, etc. The diverse set of symbiotic genes present in the different strains looked interesting. I’ve got very excited by the idea and definitely wanted to get to know if these isolated bacteria can improve the N-fixing symbiosis of the commercial strains on alfalfa, red clover and vetch, or if they could contribute to the plant development with any other way of promoting plant growth.
Though the title of the manuscript says “Increasing the legume-rhizobia symbiotic efficiency...” the data of the plant nodulation assays summarized in Tables 5 (alfalfa), 6 (vetch) and 7 (red clover) do not support this conclusion. The nodulation assays contained non-inoculated plant controls, plants inoculated only with a single bacterial strain (the corresponding commercial strain or any of the four isolates) and plants co-inoculated with the commercial strain + one of the isolates. The number of nodules, the acetylene reduction activity by a nodule or a plant, and the fresh weight of the plants were evaluated after 24-26 days kept on N-free agar medium. These categories are appropriate to measure and can say a lot about the N-fixing efficiency at the level of nitrogenase activity in the symbiosome and also at the level of plant growth and development. The whole purpose of forming N-fixing symbiosis for the plant is to get enough nitrogen compound to its development so the fresh weight of the plant should be considerably higher when they got N supply from the efficient symbiosis than when they are starving for N. And this is not the case in these experiments, there is no correlation at all between the N-fixing efficiency at the nitrogenase level and the fresh weight of the plants. This means that the conditions for the nodulation assays were not proper for the assay: the nitrogen was not the only substantial element that was missing from the medium on which the germinated plants were let grow, so another nutritive element or elements were missing and so act like a limiting factor in the growth and development of the plants. In the material and methods there is one type of medium described for all the plants and just on a quick look looks like missing e.g. Na and Cl ions, or trace element like Cu. Also, a 10 ml agar medium in a tube for two plants is very low amount for two plants to grow, especially for vetch. This means that it is not only the nitrogen which is a limiting element for the growths of these plants in this system, which can have a big influence on the overall regulation of the formation of N-fixing symbiosis. In this sense now it is not so surprising that the significantly higher fresh weight values can be found for those plants where some isolates – that do not form N-fixing nodules at all! – probably can supply the plants with some missing nutrients (alfalfa – Opo-235 or Ach-343 in Table 5; vetch – Opo-235 in Table 6; red clover – with all individual isolates in Table 7). When these isolates are co-inoculated with the respective commercial strain on the plant, there are again different outcomes: on red clover all combinations resulted in plants with significantly lower fresh weight; same on vetch and alfalfa with the above mentioned Opo-235 + commercial strain combination, while the Ach-343 + commercial strain on alfalfa resulted an even higher fresh weight, although the symbiotic efficiency was the lowest in this combination.
If we want to take into account only the N-fixing efficiency (acetylene reduction activity) values to try to figure out if the isolates made a positive contribution to that, than wee should know a little more details how these experiements were done: did you do separately the measurement by plants and by nodules, or just per plants and calculated the values for the nodules? Which nodules did you calculate and got involved in the category of NN (nodule numbers) in the Tables? Only the pink elongated nodules or some others as well? We can see from the pictures of the figures that some isolates and combinations resulted in small, white and round nodules, or even sets of nodule-like structures on the root. It would be worth to introduce these categories separately in the Tables, since those non-fixing nodules do not contribute to the N-fixation, in contrast, they are using energy from the plants. E.g. in the legend of the Tables it says: ND – not detected; but there is no ND written in these Tables, instead there is value 0 for acetylene reduction in all cases where there was no nodules, so definitely no nodules could be measured. So it is also not really possible to conclude anything about the possible contribution of the isolates to the N-fixation from these data.
In summary the isolates used in this study could promote in many cases the growth of the plants but it is not clear if they contributed to the N-fixing activity. But it’s a very exciting field of research and I encourage the authors to repeat the experiments with the proper quality and quantity medium for each plant species where the only limiting factor is nitrogen and you can keep them even longer to see the real effects. You could even use those of your strains of Opo-235 and Ach-343, which express fluorescent proteins (that you've already published) to detect them after co-inoculation.
Reviewer 4 Report
The manuscript entitled "Increasing the legume-rhizobia symbiotic efficiency due to the synergy between commercial strains and strains isolated from relict symbiotic systems" is interesting and it can be useful to improve the production of legumes in crop conditions.
The introduction is suitable to set the context of the manuscript.
The authors have explained the material and methods properly but in the last paragraph the authors have to say how many times they have carried out the nodulation assays.
Results and discussion are clear and well estructured, but there are some aspects that the authors should improve and some questions that I would like that the authors answer me:
1. Table 2: The structure of this table is not clear. The lines "Fix cluster, Nif cluster or Nod cluster" have a group of genes and in the lines below of them the authors show the presence or absence of other genes, that not always appear in the first line, so they have to change this table to do it more clear.
2. Table 5, 6 and 7: The data correspond to plant biomass, root biomass or shoot biomass has not significant differences between the control without inoculation and the plants inoculated with the comercial strain, even in case where the number of nodules increases. So why have the authors chosen this strain to inoculate plants? How do the authors explain that the improvement in number of nodules does not improve the biomass?
3. The hypothesis of the authors is that the NF produced by the isolated strains improve the nodulation of the commercial strains, but they could prove it extracting and analyzing the NFs produced by the isolates to know the composition of this pool of NFs; even adding these NFs to the plant inoculated with the commercial strain to show if the nodulation is improve too.
Reviewer 5 Report
In my opinion the manuscript can be published in Agronomy after small modifications
- I do not understand what Authors mean: "The latter isolates supplemented the commercial strains with some 20 symbiotic genes (fix, nif, nod, noe and nol) as well as the genes promoting plant growth and symbiosis 21 formation (acdRS, genes associated with the biosynthesis of gibberellins and auxins, genes of T3SS, 22 T4SS and T6SS secretion systems)." - what latter isolates, and do not other mentioned rhizobia posses nif and other genes?
- mono- and co-inoculation - what do you mean?
- "In this regard, the aim of this work was to study the effective-67 ness of symbiosis upon joint inoculation of M. varia, V. sativa and T. pratense with their 68 commercial strains and strains ....." in my opinion genum names of mentioned legumes nedds to be added, and please clarify that inoculation is prepared with bacteria but not the plants as can be assumed after this sentence seading.
- alfalfa, common vetch and red clover - please explain it immediately after usage in latin.
Round 2
Reviewer 3 Report
“ In all studied plants (alfalfa, vetch and red clover), an increase in some symbiotic parameters (number of nodules, fresh plant biomass or nitrogen fixation of nodules)”
Indeed: in “some” parameters, but the problem is - as I’ve mentioned -that these parameters are not increased in relation to each other, but many times in the opposite way, which do not support causal relationship.
“we considered white or pink formations of a certain shape (round or elongated), located mainly on the lateral roots.”
I certainly understand that technically you could not measure ARA per nodule. But you could have distinguish white and pink nodules. Or do you think that white nodules can fix N as well?
I’ve expected a more thorough revision from the authors. Besides SOME signs of elevated N-fixation that’s also quite interesting how these isolates can contribute to promoting the growth of plants. The authors did not even pay attention that e.g. when changing the ARA values per plant to ARA values per tube they should also change the calculated ARA per nodule values. Since there were two plants per tubes and in column NN they say that the nodule numbers are given per plant, then the ARA/nodule values are not correct now. Or the NN numbers are also per tube and not per plant?
Now in lines 153-154 you added: ”A suspension of each strain was prepared in the liquid medium described above”
No any liquid medium is described above.
In the previous sentence it is written: ”Seedlings were inoculated with individual strains or with a pair of strains isolated 150 from the same nodule in the amount of 106 cells of each strain per test tube”
I think it is copied from previous papers in which the authors co-inoculated plants with two isolates. But in the manuscript co-inoculation was with a commercial strain plus an isolate.
Also, would be nice to get to know more about the results of re-isolation of bacteria from nodules at the end of the test, which could help explain the results
Lines 165-166:”Strains were re-isolated from the obtained nodules and identified by 16S rDNA sequencing as described earlier”
Bacterial re-isolation was performed on individual nodules or group of nodules?
Lines 312-314:”In this regard, it is important to note that in these two variants of co-inoculation the isolates were re-isolated from nodules (the data not presented).”
Lines 374-377:”Since we did not detect an increase in nitrogen-fixing activity per nodule in variants of co-inoculation of common vetch, we do not discuss the role of nif and fix genes that are present in isolates, although the strain M. kowhaii Ach-343 was re-isolated from the nodules together with the commercial strain RCAM0626”.
That’s all information about red clover and vetch. What was the result with the other co-inoculations? What was the result in the case of alfalfa? Do the results of re-isolation of bacteria support your conclusion?
